# Microbial corrosion of DSS 2205 in an acidic chloride environment under continuous flow

Thi Thuy Tien Tran[1], Krishnan Kannoorpatti[1]*, Anna Padovan[2], Suresh Thennadil[1], Khai Nguyen[1]

1 Energy and Resources Institute, College of Engineering, Information Technology and Environment, Charles Darwin University, Darwin, Northern Territory, Australia, 2 Research Institute for Energy and Livelihood, College of Engineering, Information Technology and Environment, Charles Darwin University, Darwin, Northern Territory, Australia

☯ These authors contributed equally to this work.
* krishnan.kannoorpatti@cdu.edu.au

## Abstract

Corrosion under flow conditions is a major problem in the transportation industry. Various studies have shown the direct impact of different flow rates on bacteria biofilm formation, mass transfer and resulting different corrosion behaviour of materials in neutral environments. However, little is understood on corrosion under acidic flow conditions. This study investigated the impact of an acidic artificial seawater environment containing *Desulfovibrio vulgaris* on DSS 2205 microbial corrosion under different velocities (0.25 m.s$^{-1}$ and 0.61 m.s$^{-1}$). Experiments containing no bacteria were performed as controls. Bacterial attachment was observed by optical and scanning electron microscope (SEM). Materials corrosion was assessed using open circuit potential (OCP), electrochemical impedance spectroscopy (EIS) and potentiodynamic polarization. Pits formed after potentiodynamic test were observed under SEM. The largest area of bacterial attachment was found on coupons immersed at a velocity of 0.25 m.s$^{-1}$; however, the corrosion rate was lower than at higher velocity. Shallow pits occurred in the metal coupons when bacteria were present, while deep pits occurred in the controls. The study indicates the positive impact of biofilm formation in corrosion prevention of materials under acidic condition. The nature of corrosion behaviour of duplex stainless is discussed.

**Data Availability Statement:** All relevant data are within the manuscript and supporting informations.

## Introduction

Microbiologically influenced corrosion (MIC) or microbial corrosion refers to corrosion affected by the presence of microorganisms, which leads to the deterioration of materials such as iron, steel, concrete and stone [1–4]. MIC is a major challenge to many industries including oil, gas and mining [5]. Approximately up to 40% of all internal corrosion failure in oil pipelines are due to MIC [6–10] causing huge financial losses in the order of billions of US dollars [11]. Although there are numerous kinds of bacteria involved in accelerating the corrosion of materials, anaerobic sulphate reducing bacteria (SRB) are the chief culprits. SRB grows and

**Funding:** This research was supported by an Australian Research Training Program Scholarship provided through Charles Darwin University which award to Thi Thuy Tien Tran; grant number is 1578028. URL: https://www.cdu.edu.au/research-and-innovation/higher-degree-research/higher-degree-research-scholarships The funders had no role in study design, data collection and analysis, decision to publish, or preparation of the manuscript.

corrodes metals through redox reactions of chemical species/biogenic substance which are in direct contact with, or near, the metallic surface.

The formation of biofilms on materials' surface is the result of an uniform/nonuniform aggregation process in time or space [2] that begins immediately after immersing materials in an microbial environment. Biofilm formation on material surfaces is considered to be a crucial step to initiate microbial corrosion [12], creating a microenvironment that affects the oxidation reduction reactions and materials surface properties resulting in corrosion of materials [2]. In contrast, recent studies have shown that biofilm formed on material surfaces can be a barrier to protect and/or inhibit corrosion [2, 13]. The benefits of biofilm have recently been proposed for corrosion inhibition [14–16].

MIC can occur in stagnant or fluid flow conditions [17–19]. Although many aspects of MIC under stagnant conditions are understood, there is still some confusion regarding MIC under fluid flow conditions. Recent research showed that in stagnant and low velocity conditions, the diffusion of nutrients from the environment to the biofilm was limited, while at high velocities nutrients were replenished in the bacterial biofilm resulting in higher ATP concentrations and higher corrosion rates [20]. In contrast, it was found that biofilm development was reduced at high water velocities due to shearing of the biofilm [21]. Additionally, in the system that contains low nutrients, instead of taking electron from oxidation of carbon source, bacteria tend to take electron from the oxidation of metal to use for sulphate reduction [22]. At low flow velocity, the build-up of thick biofilm can prevent the diffusion of carbon source from environment to the bacteria under biofilm which might trigger more aggressive corrosion by SRB because of carbon starvation.

Most studies above focused on carbon steel as it is a popular material used in the pipeline transportation industry, however, there are situations where stainless steel pipes are used. Stainless steels including duplex stainless steels are reported to be susceptible to MIC and pitting [23]. Corrosion resistance of stainless steel is primarily attributed to a passive film layer of chromium oxide. For the damaged passive film to be repaired, the presence of oxygen is essential. In conditions where SRB grow, there is a lack of oxygen and the presence of biofilm and chloride ions can further reduce the repair of passive films.

Recently, we found that biofilms reduced corrosion in both acidic and alkaline conditions and that the microbes modified the pH of alkaline and acidic environments to a near neutral pH of 7.3 [24, 25]. It appears that pH 7.3 is the preferred pH for *D. vulgaris*. These results were based on experiments performed in static conditions, and it was unclear as to how the microbes would behave under flow conditions in acidic environments. To address this, a study was undertaken to assess MIC of duplex stainless steel 2205 at pH 5 under different flow conditions. Coupons were incubated with anaerobic bacteria (*D. vulgaris*) at velocities of 0.25 m.s$^{-1}$ and 0.61 m.s$^{-1}$. An incubation containing no bacteria was used as a control. The corrosion behaviour under these conditions was studied by electrochemical testing including measuring open circuit potential (OCP), electrochemical impedance spectroscopy (EIS) and potentiodynamic polarization. Biofilm, corrosion products and pitting morphology were observed and analysed by optical microscope, scanning electron microscope (SEM) and energy-dispersive X-ray spectroscopy (EDS).

## Materials and methods

### Materials

The experiment was performed using seawater at two velocities (0.25 m.s$^{-1}$ and 0.61 m.s$^{-1}$), each with and without *D. vulgaris* (4 treatments). Two DSS 2205 coupons (10 mm x 10 mm x 2 mm) were used for each treatment. The chemical composition of the coupons, analysed by

energy dispersive X-ray fluorescence spectrometry (EDX-8100), was in wt.%: Mn 1.68, S 0.053 Si 0.43, Cr 22.1, Ni 6.12, Mo 2.89 and balanced Fe. One coupon was connected to the potentionstat and acted as working electrode. Another coupon was used for studying biofilm formation.

All coupons were polished to 1 μm diamond-paste surface finish then followed by rinsing with distilled water, degreased with acetone, rinsed again with distilled water. The coupons were eliminated of any bacteria contamination prior to the experiments by immersing in 80% (v/v) ethanol for 2 hours to and finally were dried in a biohazard cabinet.

## Medium and bacterial culture

*D. vulgaris* (ATCC® 7757) (In Vitro Technologies, VIC) was retrieved from -80°C glycerol stock and cultured in modified Baar's medium under anaerobic conditions. *D. vulgaris* was advised to growth at optimum condition of 37°C of temperature, pH 7.4 and in modified Baar's medium by the provider. Previous studies show that *Desulfovibrio* sp. prefer to grow at pH range 6 to 8 [26]. In this study, the environmental pH of 5 was used to create acidic condition.

The medium and bacterial culture was prepared as described elsewhere [25]. In brief, nutrient rich artificial seawater (test solution) which consisted of modified Baar's medium in artificial seawater. Media was adjusted to pH 5 with 1 M HCl. The tested solution was purged with nitrogen gas for 1 hour to remove air and autoclaved for 15 mins at 121°C for sterilizing.

The bacterial culture was added to the test solution (5 L total) and the final bacterial concentration was approximately $3.17 \times 10^4$ cells mL$^{-1}$. The control solution consisted of nutrient rich artificial seawater without the bacteria.

Glutaraldehyde (2.5%) was prepared for staining bacterial biofilms prior to surface analyses and phosphate-buffered saline 1X (PBS) was prepared for samples preparation before doing surface analysis.

The experiments were operated at 37°C for 14 days which is the optimum temperature bacteria.

## Flow circulation system and test condition

The lab-scale flow circulation system (S1 Fig) consists of a 6-litre tank connected to the tested cell by clear vinyl tube, tank outlet valve, pump, flow indicator and the tested cell. The rectangular tested cell includes a reference electrode Ag/AgCl, a platinum coated counter electrode and two side holes for attaching two tested coupons. One coupon was connected to a potentiostat as working electrode and a second coupon was used for biofilm formation. The cell was constructed from acrylic glass in order to monitor inside the cell. Tested samples were attached to the sides of the cell and the area exposed to fluid was approximately 1 cm$^2$.

Coupons were attached to the rectangular test cell. Nitrogen gas was introduced to the system through a hole on the top of the tank and flushed for 20 minutes to remove air inside the system. The tank was filled with 5 litres of nutrient rich seawater (with or without *D. vulgaris* depending on the treatment) along with the flow of nitrogen gas to prevent the introduction of air into the system. After filling, a layer of nitrogen gas was applied above the medium in the tank to create anaerobic environment inside. Finally, the hole on the top of the tank was covered with epoxy. The medium was circulated in the loop system by a pump. The total solution volume including in the tank and pipe was 5 litres. Flow rate was adjusted using the outlet tank valve and was monitored using a flow indicator. The size of the rectangular cell was 44 mm x 44 mm.

The systems were operated at velocities of 0.25 m.s$^{-1}$ and 0.61 m.s$^{-1}$. The velocity of 0.25 m.s$^{-1}$ was chosen which is same as previous study [27]. The velocity of 0.61 m.s$^{-1}$ was the maximum that the pump capacity. The calculated Reynolds (Re) numbers were 1.1 x 10$^4$ at 0.25 m.s$^{-1}$ and 2.67 x 10$^4$ at 0.61 m.s$^{-1}$ The Re numbers show the turbulent condition of the flow through the attached coupons.

Each experiment was carried out for a total of 320 hours at room temperature (26˚C). The time was determined based on literature showing that after approximately 300 hours, bacteria gradually die due to the lack of nutrient source such as sulphate at pH 5 [24].

## Analytical methods

All the electrochemical tests were taken during continuous flow conditions.

Variation of pH during experimental time.

The pH of the medium in the tank was monitored by removing 5 mL through the epoxy hole using a syringe with needle. After taking the solution, the hole was covered with a new epoxy layer. The solution was then filtered using a 0.45 μm filter to eliminate the bacteria and was the pH of the filtrate measured.

**Open circuit cotential (OCP).** A platinum coated electrode and an Ag/AgCl electrode was used as counter electrode and a reference electrode, respectively. The electrochemical experiments were performed using Gamry Reference 3000 and the results were analysed using Gamry Echem Analysis software. OCP test was recorded daily for each specimen.

**Electrochemical impedance spectroscopy (EIS).** EIS was recorded every 20 hours. The tests were performed at OCP value using 10 mV amplitude and the frequency range from 0.05 Hz to 100000 Hz. The results were analysed by Gamry Echem Analysis software using a suitable equivalent circuit model.

**Potentiodynamic polarization.** A scan rate of 0.5 mVs$^{-1}$ was used in potentiodynamic polarization which started from -0.25 V vs. OCP to transpassive potential test after 320 hours exposure. The corrosion potential ($E_{corr}$), corrosion current density ($I_{corr}$) and pitting corrosion ($E_{pit}$) were obtained from the polarization curves by using software Gamry Echem Analysis.

**Surface analysis.** The coupons used for biofilm formation were observed by scanning electron microscopy (SEM) (Jeol Model JSM 5610LV) before each experiment to ensure the surfaces were uniform. After 320 hours, these coupons were removed from the media and dried at 600C. The coupons were then analysed with an optical microscope (Nikon Eclipse MA 100) and SEM for observing biofilm formation, and energy-dispersive X-ray (EDX).

To detect pits formed on the coupon surfaces after the potentiodynamic test, the corrosion products along with biofilm formed on coupons' surface were removed. The procedure for preparing coupons for observing pitting is described elsewhere [28]. Briefly, the coupons were washed prior to immersing in Clarke's solution. Finally, they were rinsed with high pure water followed by 80% (v/v) ethanol and dried. The coupons' surface was observed and photographed using a stereo microscope (Olympus SZ40) for overall pitting formation on the surface. The photos were analysed for total pit formation and corroded areas using Image J software. Pit depth was measured using an Elcometer E224-TI. The coupons were then used for surface analysis by SEM for pitting morphology.

## Results

### pH variation

The pH of the media containing bacteria increased significantly from 5 to approximately 7.6 at both velocities, whereas there was only a slightly increase from pH 5 to around 5.2 in the

controls at both low and high velocity conditions, Fig 1(A). The significant change in pH in the biotic environment can be explained by bacterial metabolism and is the same as occurred in stagnant condition [24]. The slightly increase in pH in the control experiment can be due to the loss of proton to dissolving of passive film and corrosion.

**Open circuit potential (OCP).** Fig 1(B) shows the unstable OCP values (vs Ag/AgCl) irrespective of treatment. The changes in OCP in the bacterial environment can be attributed to corrosion [29] and presence of the biofilm [25]. Generally, the coupons in biotic conditions had a more negative shift than coupons in control conditions which can be due to the presence of biofilm formed on the surface. In both biotic and control conditions, OCP values of high velocity were higher than low velocity which indicates that coupons in high velocity were more prone to corrosion than at low velocity.

## Biofilm formation

Photos of the biofilm formed on the surface of the coupons in different conditions taken by optical microscope and SEM are shown in Figs 2 and 3 respectively. It can be clearly seen in Fig 2 that at the lower velocity of 0.25 m.s$^{-1}$, the area of biofilm on the coupons was greater than at the higher velocity 0.61 m.s$^{-1}$. It is possible that the either the biofilm did not adhere sufficiently or was eroded under high velocity conditions. This result is similar to previous studies [27, 30, 31]. The EDX spectrum showed higher sulphide levels on coupons at velocity of 0.25 m.s$^{-1}$ than at 0.61 m.s$^{-1}$. Higher bacterial attachment might result in higher biogenic sulphide production due to their metabolism. This could be another line of evidence that the biofilm formed at low velocity was thicker than at high velocity.

## Electrochemical impedance

Nyquist plots and Bode plots obtained from EIS test for each treatment are shown in Figs 4 and 5 and EIS parameters were shown in Table 1. The data fitted the electrical equivalent circuit for control and biotic conditions (S2 Fig). The EEC model $R_s[Q_{CPE}R_{ct}]$ was used for the control environment and the model $R_s[Q_{CPE}[R_b[C_{dl}R_{ct}]]]$ was used for microbial environment [32, 33]. The parameters included in the model are: $R_s$: solution resistance, $R_b$: biofilm/passive film resistance, $R_{ct}$: charge transfer resistance, $C_{dl}$: electrical double layer capacitance and a constant phase element (CPE). The Nyquist plots show a capacitive arc which represents film resistance of the materials. A large radius of the capacitive arc indicates high resistance of the materials to corrosion.

It can be clearly seen in the control experiment that the coupons at higher velocity (0.61 m.s$^{-1}$) had generally lower radius of capacitive arcs than at low velocity (0.25 m.s$^{-1}$), thus lower in impedance during 320 hours immersion (Fig 4(A) and 4(B)). During the time of immersion, the charge transfer resistance ($R_{ct}$) of the coupons at low velocity was higher than at high velocity (Table 1). This can result in weaker corrosion resistance of DSS 2205 in high velocity acidic control environment than in low velocity. This can be seen in Bode plots in Fig 5(A) and 5(B). The phase angle of low velocity coupon had wide range of frequency where the phase close to 90˚ while in high velocity, the range was narrower. The wider range, the better passive film formation on the surface, thus the better corrosion resistance. The weak corrosion resistance of coupons at high velocity can be due to the high diffusion of proton from environment to materials surface and erosion caused by high velocity. The $R_{ct}$ values of both samples decreased with immersion times (from 6. 26 x 10$^2$ kΩcm$^2$ to 1.41 x 10$^2$ kΩcm$^2$ at low velocity condition and from 1.65 x 10$^2$ kΩcm$^2$ to 0.73 x 10$^2$ kΩcm$^2$ at high velocity condition) as the passive was damaged and corrosion occurred.

(A)

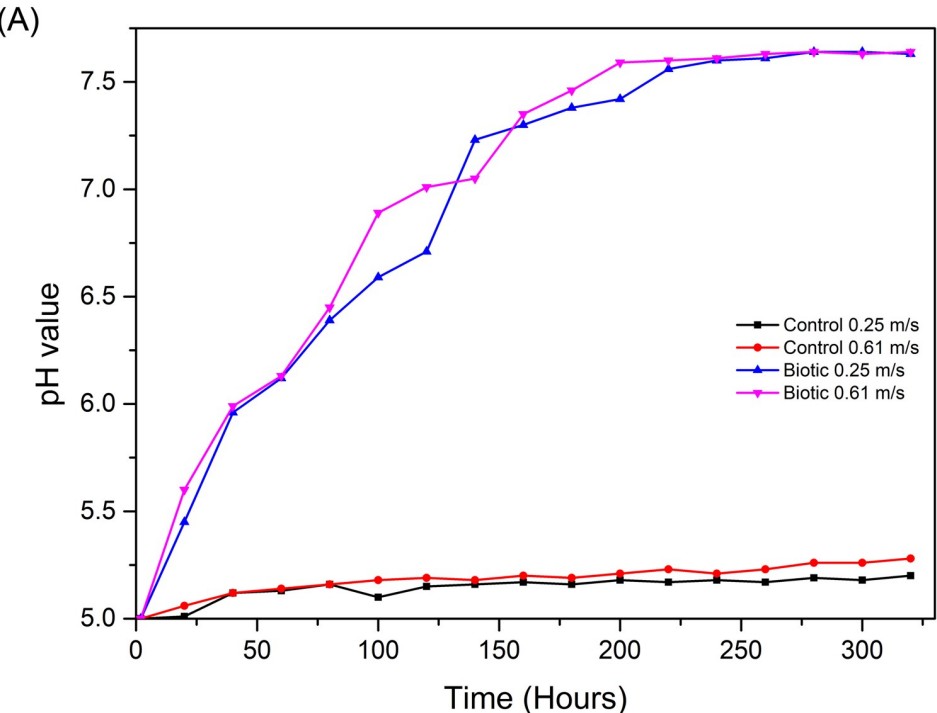

(B)

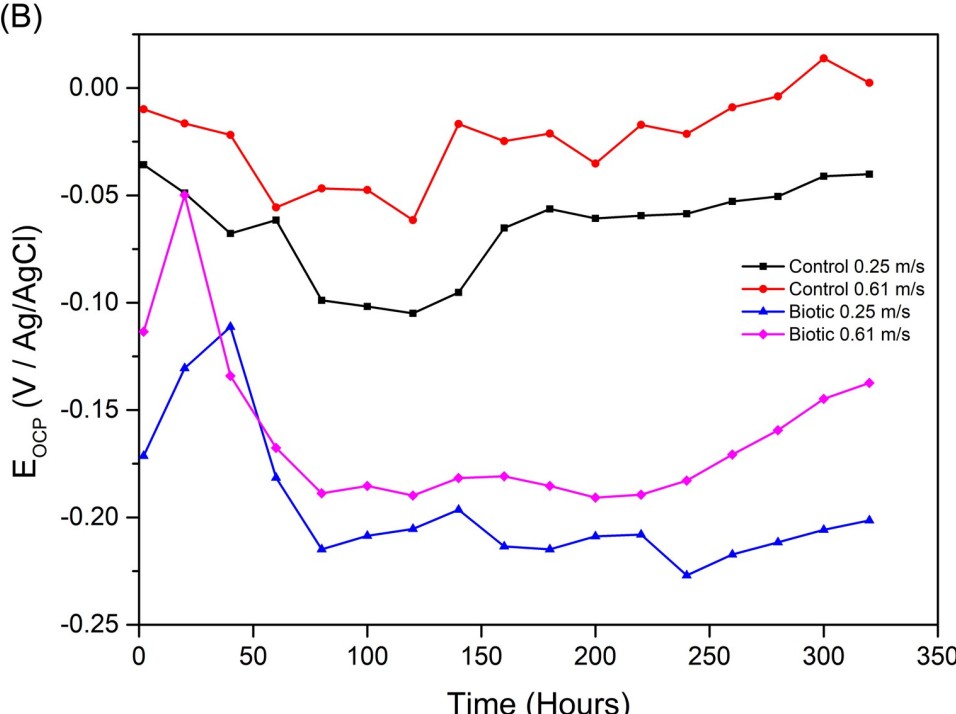

**Fig 1. (A) Changing of environmental pH during 320 hours exposure in different conditions (B) OCP decay curve of each coupon in control and biotic environments at different velocities.**

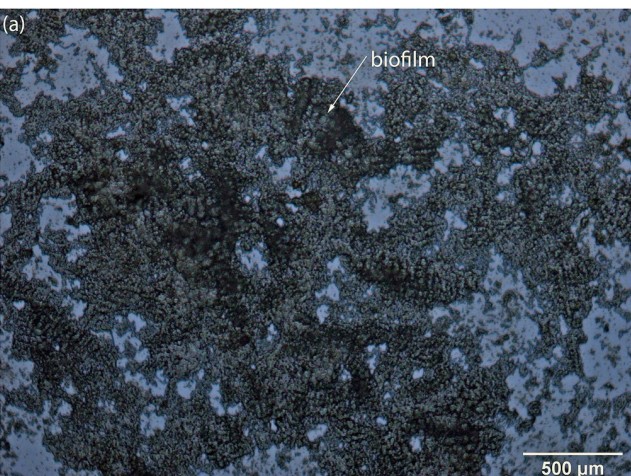
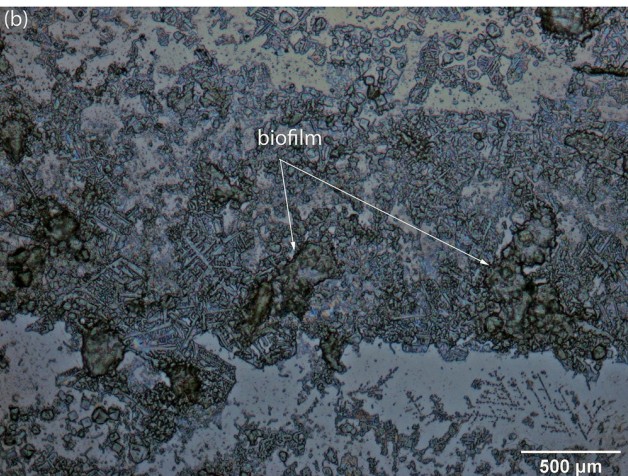

**Fig 2. Optical images of biofilm formation on coupons at velocity of 0.25 m/s (A) and 0.61 m/s (B).**

In the microbial environment, generally the radius of capacitive arcs of the coupons two different velocities showed no difference in magnitude (Fig 4(C) and 4(D)). It is interesting to notice that there were three stages of impedance behaviour of the coupons during exposure time in both low velocity and high velocity including the change from kinetic control to mass transfer control within around 20 hours of immersion, the increase and decrease in impedance of the coupons. During the first 2 hours, the capacitive arcs of both coupons were curved shape which is similar to the control condition which presents the kinetic control. However, the shape of the capacitive arc changed to almost a straight line at low velocity after 20 hours and after 2 hours at high velocity. This presents the change from kinetic control to diffusion region where mass transfer is significant which can imply the growth of the biofilm on the coupons. This change can also be seen in Bode plots (Fig 5(C) and 5(D)) where the phase curve at very low frequency increased to around 80˚ after 2 hours immersion in coupon at low velocity and 20 hours for coupon at high velocity. After the changing stage, the impedance of the coupons increased and then remained constant for approximately 140 hours from 20 hours to 160 hours for low velocity and approximately 100 hours from 20 hours to 120 hours for high velocity. The polarisation resistance ($R_p$) of the coupons can be determined by the total resistance of charge transfer resistance ($R_{ct}$) and biofilm/passive film resistance ($R_b$). It can be seen in Table 1 that $R_p$ of the coupons at low velocity increased with exposure time before decreasing after 160 hours immersion from $1.26 \times 10^2$ k$\Omega$cm$^2$ to $4.35 \times 10^2$ k$\Omega$cm$^2$. At high velocity, it increased from $1.88 \times 10^2$ k$\Omega$cm$^2$ to $1.93 \times 10^2$ k$\Omega$cm$^2$. This indicates the growth of the biofilm on the surface of materials increased corrosion resistance of the coupons as the biofilm acted as a barrier to prevent the diffusion of protons from environment to the material surfaces. Additionally, thicker of biofilm was formed on the coupon at low velocity than at high velocity. The impedance then decreased after 160 hours of immersion for low velocity and after 120 hours of immersion for high velocity. This is same for the phase in Bode plots (Fig 5(C) and 5(D)) where the phase decreased after same times as impedance. Higher phase at very low frequency indicates higher capacitance. The decrease in impedance and phase suggests the start of the corrosion process and that it occurred at the interface between solution and the biofilm/passive film. At high velocity, the biofilm was less dense with low coverage of the coupon surface, resulting in a higher diffusion rate of protons to the material surface than at low velocity. The presence of protons and chloride ions from the environment caused damage to the passive

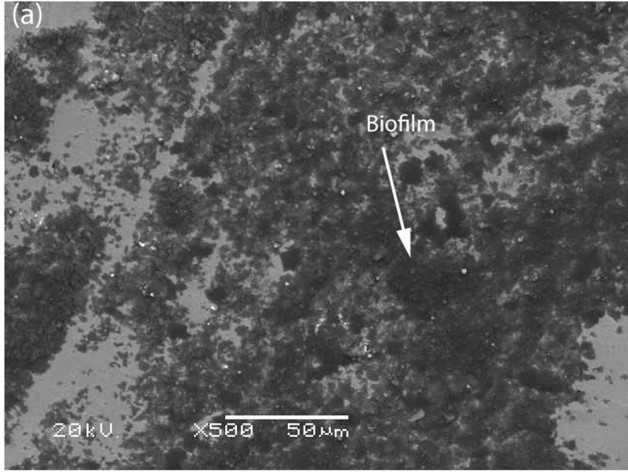

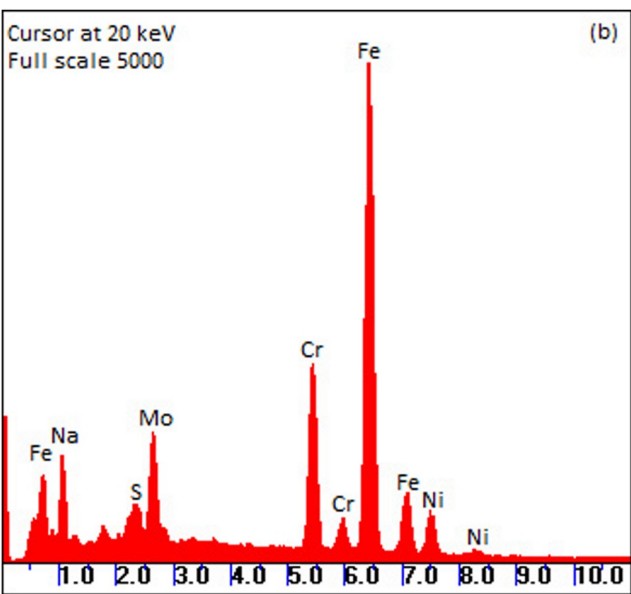

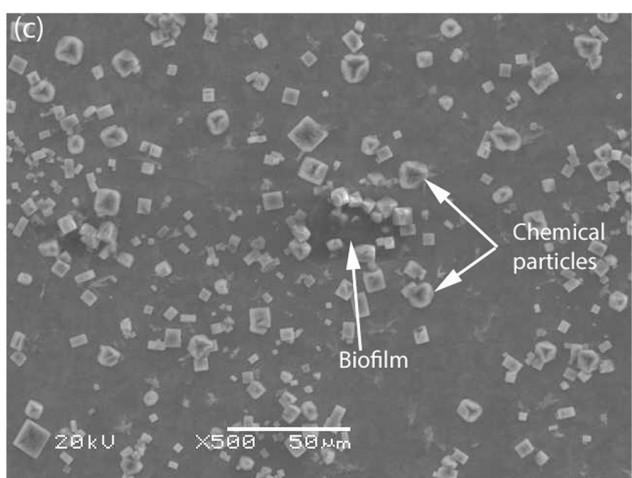

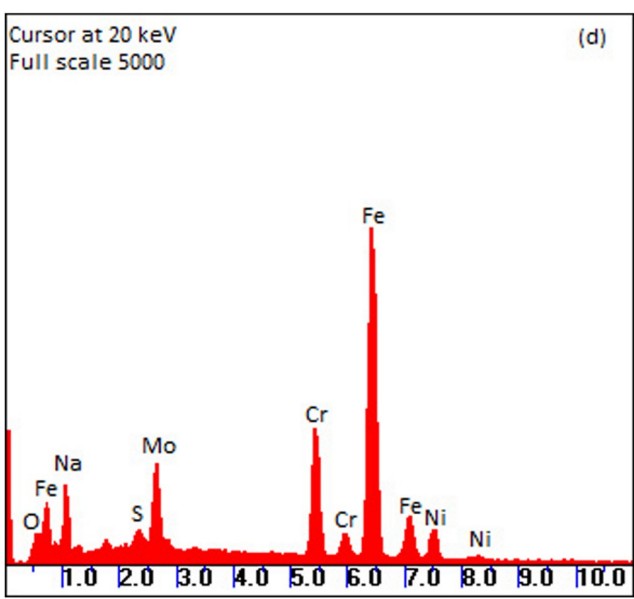

**Fig 3. SEM-EDX images of biofilm formation on coupons at velocity of 0.25 m/s (A, B) and 0.61 m/s (C, D).**

film and resulted in corrosion. Thus, the decrease in impedance of coupons at high velocity was faster than at low velocity.

## Polarization curves

Fig 6 shows the polarization curves of coupons in both control and biotic conditions at velocity of 0.25 m.s$^{-1}$ and 0.61 m.s$^{-1}$ and important parameters are shown in Table 2.

Generally, the corrosion rate of the coupons at high velocity was higher than at low velocity in both control and biotic conditions. This is in good agreement with electrochemical impedance results. It is interesting to notice that, at the same velocity, coupons in the biotic condition had a lower corrosion rate than in control conditions. Additionally, pitting corrosion

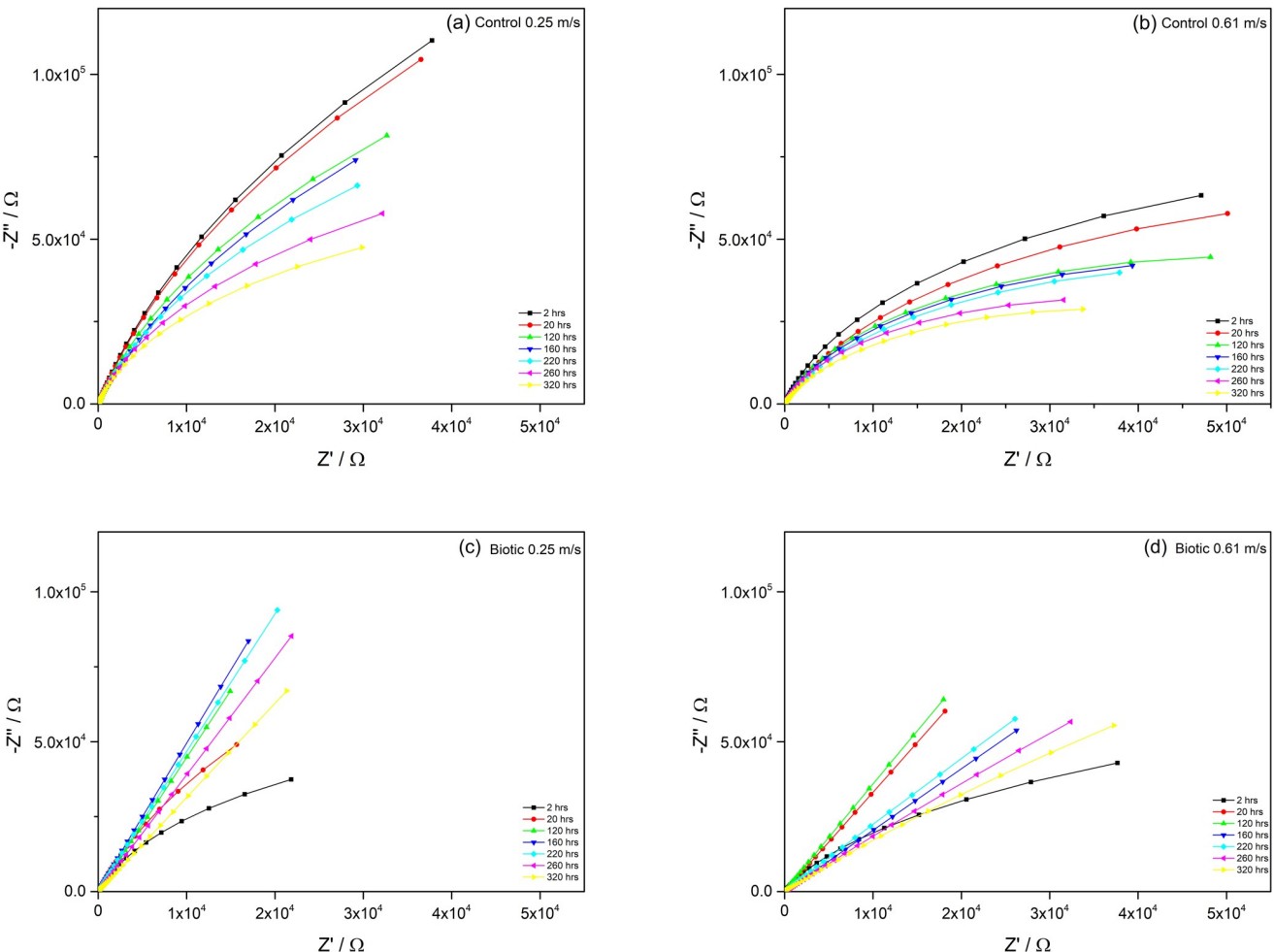

**Fig 4.** Nyquist plots of coupons immersed in control condition (A) at 0.25 m/s. (B) 0.61 m/s and biotic condition (C) 0.25 m/s, (D) 0.61 m/s during 320 hours immersion.

potentials of coupons in biotic environment were higher than in control condition which suggests better corrosion resistance of coupons in biotic conditions. This can be explained by the formation of the biofilm on the surface of the coupons which decreased the diffusion of aggressive species such as protons and chloride ions from the environment to the surface. Furthermore, the biogenic sulphide produced by SRB metabolism can bind with metal ions to form corrosion products in the biofilm matrix which can act as a temporary protective layer to prevent corrosion. Sulphide present in the biofilm and corrosion products matrix can be seen in Fig 3(B) and 3(D).

## Quantification of pits and morphology

Fig 7 shows the surface of the coupons and these images were used to calculate the number of pits, the area of corrosion and the average pit depth (Table 3). The number of pits were counted over an area $> 0.1$ mm$^2$.

Generally, at the same velocity, the total corroded area in control conditions was lower than in biotic conditions, however, the average pit depth was higher. Thus, pits in control

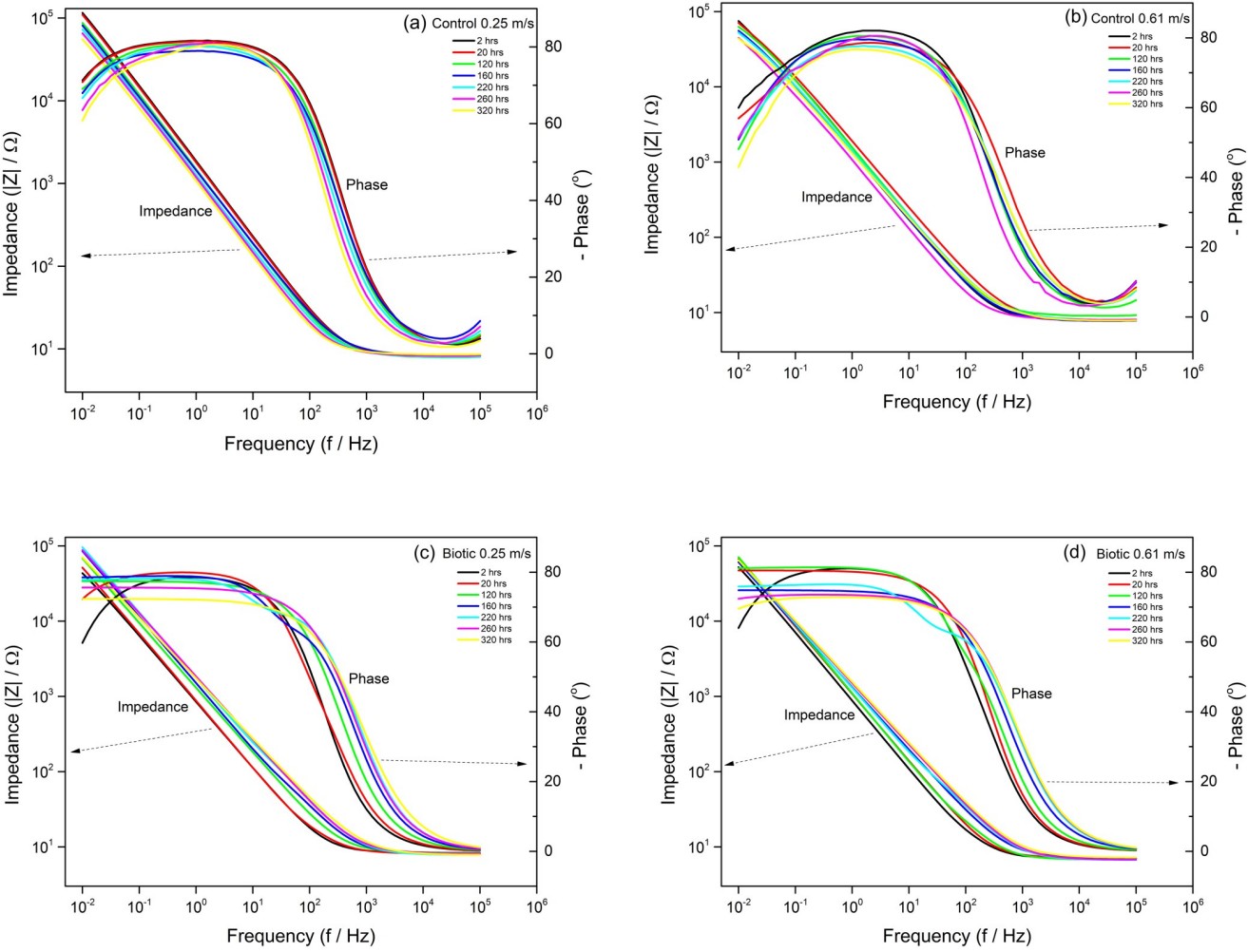

**Fig 5.** Bode plots of coupons immersed in control condition (A) at 0.25 m/s. (B) 0.61 m/s and biotic condition (C) 0.25 m/s, (D) 0.61 m/s during 320 hours immersion.

conditions were narrow and deep (Fig 7(B) and 7(C)) while pits in the biotic conditions were wider and shallower (Fig 7(D) and 7(E)).

## Discussion

MIC is a growing problem in various industries and results in huge financial losses. The initial step toward corrosion is the attachment of bacteria to the material surface and the biofilm formation. Biofilm contains > 95% water and a matrix of exopolysaccharide substances (EPS) [34]. The mechanism of corrosion enhancement caused by biofilm can include [2]: (i) Biofilm affects the transportation of chemical species from or towards the materials' surface; (ii) The detachment of the biofilm facilitates the removal of the protective films or the formation of biofilm changes the passive film structure and/or increases the dissolution of the passive film; (iii) Patchy distribution of the biofilm causes differential aeration effects on materials' surface and oxidation–reduction conditions changes at the metal–solution interface.

Sulphate reducing bacteria (SRB) are one of the notorious predominant group of bacteria involved in MIC [35, 36] and they are widespread. SRB catalyses sulphate ions to sulphides

**Table 1. Electrochemical model impedance parameters of DSS 2205 in different conditions.**

| | | Parameters | 2 hours | 20 hours | 120 hours | 160 hours | 220 hours | 260 hours | 320 hours |
|---|---|---|---|---|---|---|---|---|---|
| **Abiotic** | **0.25 m.s⁻¹** | $R_s$ ($\Omega cm^2$) | 8.2 | 8.2 | 8.1 | 8.4 | 8.2 | 8.4 | 8.6 |
| | | $Q_{CPE}$ ($\mu Fcm^{-2}s^n$) | 10.1 | 10.6 | 12.9 | 13.6 | 15.2 | 16.2 | 18.4 |
| | | n | 0.91 | 0.91 | 0.89 | 0.88 | 0.89 | 0.9 | 0.89 |
| | | $R_{ct}$ ($k\Omega cm^2$) x $10^2$ | 1.73 | 1.15 | 1.12 | 1.11 | 1.03 | 0.96 | 0.94 |
| | **0.61 m.s⁻¹** | $R_s$ ($\Omega cm^2$) | 7.9 | 7.9 | 8.2 | 7.9 | 8.1 | 8.1 | 7.9 |
| | | $Q_{CPE}$ ($\mu Fcm^{-2}s^n$) | 12.6 | 11.1 | 12.5 | 14.8 | 14.6 | 19 | 15.9 |
| | | n | 0.9 | 0.86 | 0.89 | 0.88 | 0.86 | 0.89 | 0.85 |
| | | $R_{ct}$ ($k\Omega cm^2$) x $10^2$ | 1.65 | 1.55 | 1.06 | 1.05 | 1.04 | 0.76 | 0.73 |
| **Biotic** | **0.25 m.s⁻¹** | $R_s$ ($\Omega cm^2$) | 8.4 | 8.1 | 8.3 | 8.1 | 7.8 | 8.2 | 7.8 |
| | | $Q_{CPE}$ ($\mu Fcm^{-2}s^n$) | 23.3 | 19.7 | 15.8 | 11.6 | 10.3 | 11.6 | 13.2 |
| | | n | 0.88 | 0.88 | 0.86 | 0.86 | 0.85 | 0.84 | 0.84 |
| | | $R_b$ ($\Omega cm^2$) | 67.7 | 69.7 | 76.1 | 81.5 | 77.9 | 70.2 | 67.3 |
| | | $C_{dl}$ ($\mu Fcm^{-2}$) | 2.39 | 1.74 | 1.65 | 1.55 | 1.63 | 1.74 | 1.81 |
| | | $R_{ct}$ ($k\Omega cm^2$) x $10^2$ | 1.26 | 3.77 | 4.28 | 4.35 | 3.96 | 3.91 | 3.51 |
| | **0.61 m.s⁻¹** | $R_s$ ($\Omega cm^2$) | 7.7 | 7.9 | 7.4 | 7.7 | 7.8 | 7.3 | 7.4 |
| | | $Q_{CPE}$ ($\mu Fcm^{-2}s^n$) | 17.9 | 16.2 | 13.5 | 16.9 | 14.9 | 12.9 | 13.4 |
| | | n | 0.89 | 0.89 | 0.89 | 0.84 | 0.83 | 0.83 | 0.82 |
| | | $R_b$ ($\Omega cm^2$) | 87.2 | 91.3 | 92.6 | 74.1 | 78.9 | 71.6 | 69.6 |
| | | $C_{dl}$ ($\mu Fcm^{-2}$) | 1.68 | 1.56 | 1.55 | 2.04 | 1.99 | 0.201 | 2.09 |
| | | $R_{ct}$ ($k\Omega cm^2$) x $10^2$ | 1.88 | 1.92 | 1.93 | 1.75 | 1.78 | 1.71 | 1.68 |

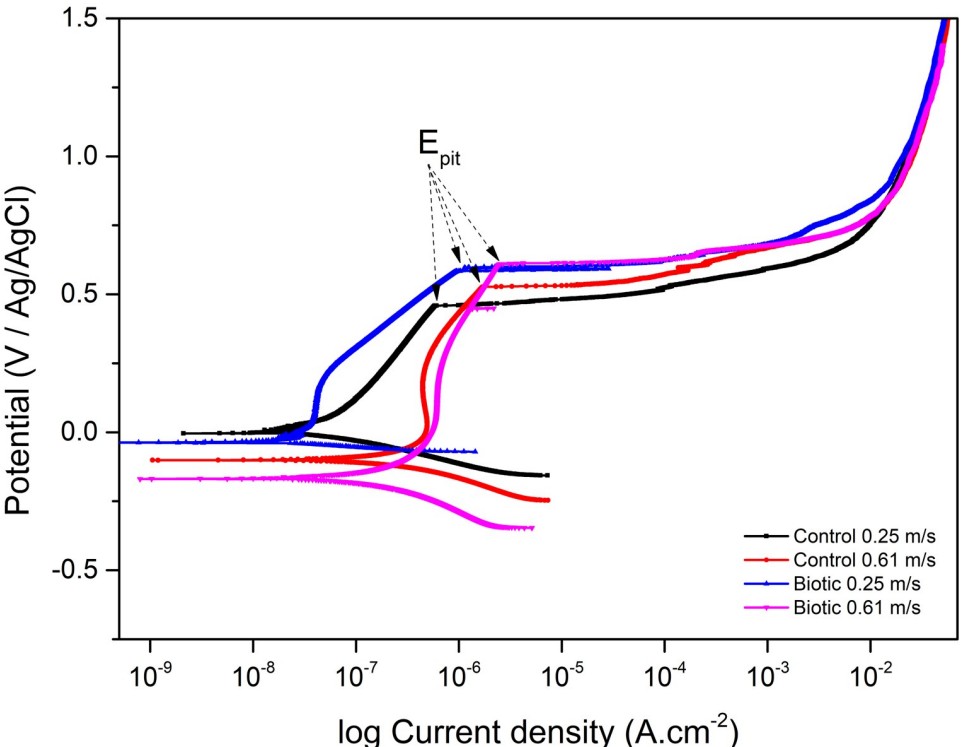

**Fig 6. Polarization curves of coupons immersed in different conditions.**

**Table 2. Polarization parameters of coupons immersed in different conditions.**

| Parameter | Control 0.25 | Control 0.61 | Biotic 0.25 | Biotic 0.61 |
|---|---|---|---|---|
| | (m.s$^{-1}$) | | | |
| $i_{corr}$ (nA) | 52.7 | 733 | 25.8 | 543 |
| $E_{corr}$ (mV) | -2.41 | -100.2 | -37.4 | -201.4 |
| Corrosion rate (mmpy) | $6.79.10{-}3$ | $9.44.10{-}2$ | $3.32.10{-}3$ | $6.99.10{-}2$ |
| $E_{pit}$ (V) | 0.459 | 0.528 | 0.585 | 0.611 |

through anaerobic respiration and corrodes materials as a consequence of oxidation and reduction reactions [37, 38]. The anodic reaction and anodic reactions can be expressed as follows [22, 39]:

Oxidation reactions:

$$4Fe \rightarrow 4Fe^{2+} + 8e^- \tag{1}$$

$$CH_3CHOHCOO^- + H_2O \rightarrow CH_3COO^- + CO_2 + 4H^+ + 4e^- \tag{2}$$

Reduction reactions:

$$SO_4^{2-} + 9H^+ + 8e^- \rightarrow HS^- + 4H_2O \tag{3}$$

At the surface of the passive film, the presence of sulphide and chloride ions can competitively adsorb with -OH [40] resulting in passive film break down. In this study, the decreasing impedance and phase indicates microbial corrosion occurred during time of exposure. However, the corrosion rate of materials in microbial environment was lower than in control conditions.

The consumption of protons (Eq 3) can explain the rise in pH in the biotic treatment and this can be one of the factors that contribute to a reduced corrosion rate in biotic conditions compared to control conditions due to the decrease of proton concentration in the biotic environment.

In the absence of bacteria, and in the presence of high proton concentrations, the passive film of DSS 2205 can be dissolved according to equations 5 to 7 [41]. The slight increase of pH in the absence of bacteria can be attributed by the consumption of protons according to the following equations.

$$Fe_2O_3 + 12H_2O + 6H^+ + 2e^- \rightarrow 2Fe(H_2O)_6^{2+} + 3H_2O \tag{4}$$

$$3Cr_2O_3 + 10H^+ \rightarrow 2Cr_3(OH)_4^{5+} + H_2O \tag{5}$$

$$Cr_2O_3 + H_2O + 2H^+ \rightarrow 2Cr(OH)_2^+ \tag{6}$$

Therefore, the dissolution of the passive film can lead to a positively charged layer which significantly affects the absorption of aggressive anions such as chloride to migrate to the film surface. Along with protons, chloride attacks the passive film causing deep pits in the material surface. The high velocity, the high mass transfer of such ions to material surface and the high shear stress on the passive film of material, thus resulted in a higher corrosion rate. This was the same as biotic conditions where higher velocity resulted in a higher corrosion rate. However, in comparison with the control experiment at the same velocity, the corrosion rate of materials in the microbial environment was lower.

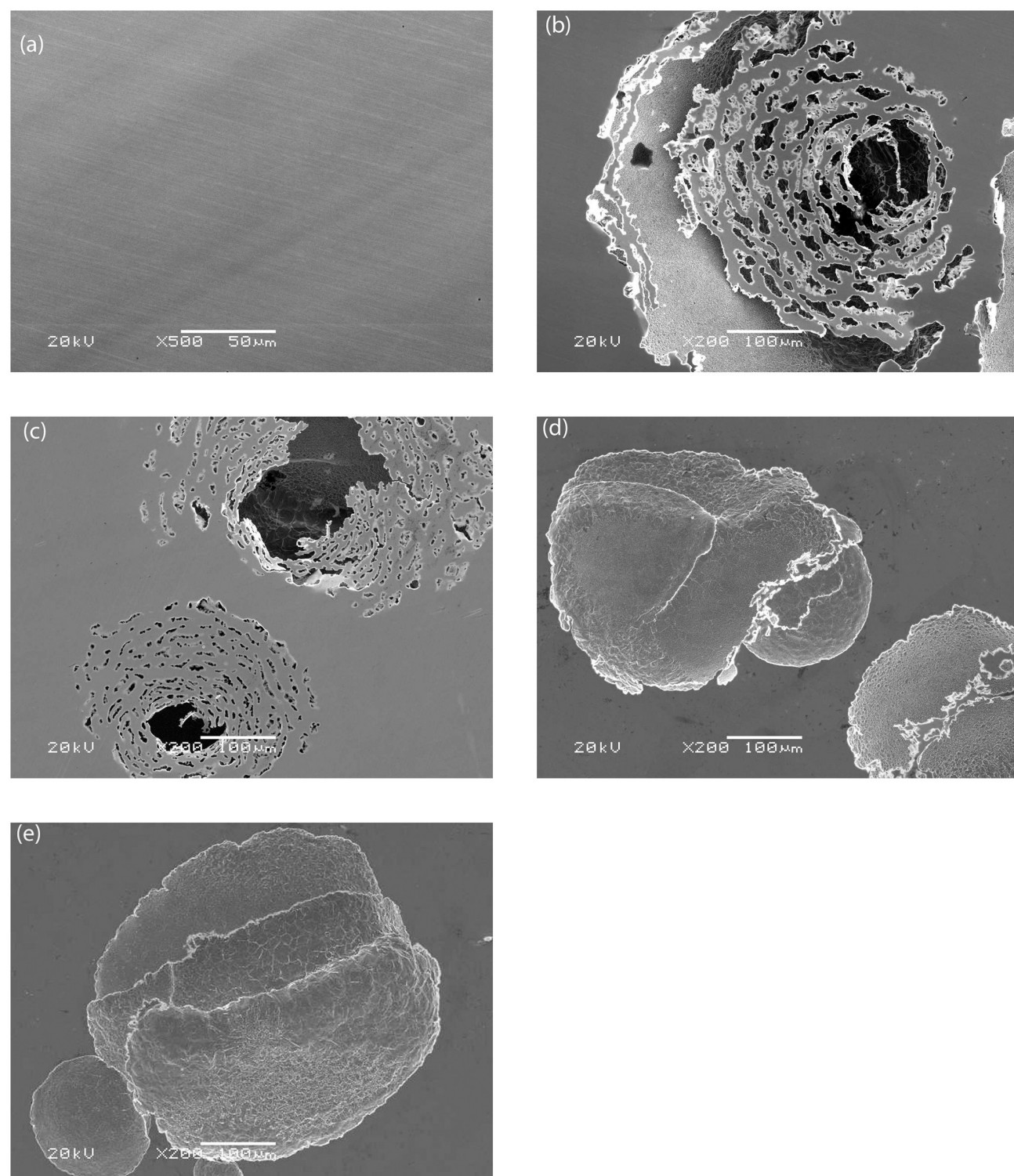

**Fig 7. Pit morphology (A) before experiment, control condition (B) 0.25 m/s, (C) 0.61 m/s and biotic condition (D) 0.25 m/s, (E) 9.61 m/s.**

**Table 3. Pits quantification parameters of each coupons in different test condition.**

| Conditions | Total number of pits | Corroded area (mm$^2$) | Average pit depth (μm) |
|---|---|---|---|
| Control 0.25 m.s$^{-1}$ | 122 + 13 * | 152.36 + 26.12 | 120.21 +9.4 |
| Control 0.61 m.s$^{-1}$ | 74 + 6 | 107.94 + 23.16 | 148.52 + 9.6 |
| Biotic 0.25 m.s$^{-1}$ | 146 + 11 | 188.93 + 29.23 | 48.63–4.69 |
| Biotic 0. m.s$^{-1}$ | 97 + 5 | 293.57–24.5 | 96.71–10.12 |

*: The total values presented the second values by repeated the experiment.

Generally, the corrosion rate of the tested coupons can be summarised in order of high to low corrosion rate: control high flow > biotic high flow > control low flow > biotic low flow. The difference in the corrosion rate of the materials in the different treatments trialled in the present study can be explained by two reasons, (i) biofilm formation and (ii) presence of aggressive ions.

Previous studies have shown that at low velocity or in stagnant conditions, bacteria tend to form thicker biofilms on material surfaces due to less shear stress compared to high flow rates. This results in high corrosion rates [27, 42] due to bacterial corrosion (Eqs 1–3) and the presence of sulphide ions which are corrosive and harmful to the passive film. Additionally, at low velocity or in stagnant conditions, the biofilm build up and the diffusion of nutrients from the environment to the bacteria in the biofilm matrix is limited, results in nutrient deprivation [20], e.g carbon source. Dake et. al. [22] stated that under carbon source starvation, *D. vulgaris* could switch from carbon source oxidation energy (Eq 2) to mostly iron oxidation energy (Eq 1), thus causing severe corrosion under biofilm conditions. However, the authors did not discuss sulphate ions which are also a nutrient and can act as a bacterial electron acceptor terminator for *D. vulgaris* respiration. A lack of sulphate ions might result in reducing corrosion reactions and producing less biogenic sulphide which is one of the main reasons for accelerating corrosion rates [38]. Thus, when both sulphate ions and carbon sources are lacking, e.g low diffusion of nutrient from environment to bacteria in biofilm matrix, the build-up of biofilm might not accelerate corrosion. Also, all those studies were done at near neutral pH where H$^+$ ions are depleted. A previous study showed that in low pH stagnant condition, biofilm can be protecting the metal from corrosion [24].

In the present study, these observations were challenged, with lower corrosion rates observed at low velocity (0.25 m.s$^{-1}$) and higher corrosion observed at higher velocity (0.61 m.s$^{-1}$). This can be explained by the presence of protons in the acidic conditions and aggressive ions such as chloride in the seawater. As the passive film was dissolved according to Eqs 4–6, the metal was more easily corroded by protons and chloride ions. At low velocity, due to the presence of biofilm formed on the material surface, the diffusion of aggressive ions was limited, thus decreasing the attack by those ions and reducing corrosion. In other words, the biofilm acted as a protective layer and resulted in a decreased corrosion rate. At higher velocity, the biofilm was either not adhere sufficiently or was eroded under high velocity conditions eroded, thus resulting in around 2 times higher corrosion rate (Table 2). This can be compared between biotic and control conditions. In biotic conditions, the biofilm played an important role in corrosion prevention in this study thus lowering the corrosion rate of the tested materials compared to control conditions.

The shape of the pits was different in coupons exposed to bacteria compared to control conditions. Pits were narrow and deep in the coupons without bacteria, but wide and shallow pits formed in the presence of bacteria. This reveals that in biotic conditions, the binding of metal ions and sulphide can act as a protective film on the surface of materials along with the biofilm

to prevent proton and chloride attack. In contrast, in the control experiments, there was no such protective film to prevent corrosion of the material. When the passive film was ruptured, an electrochemical cell became active. Metal, e.g. iron in the more anodic bottom of the pit dissolved into the solution. The concentration of the chloride ions in a pit could increase as the pit deepens, thus the consequences were accelerated pitting creating narrow deep pits. In industrial systems, the narrow deep pits are worse than wide and shallow pits as it can cause serious failures of an entire engineering systems such as leakage.

This study has revealed an important observation which is useful for corrosion prevention and protection in the fluid transportation industry. In acidic conditions, to prevent microbial corrosion, flow should occur at as low velocity as possible. Traditionally, fluid flow should be transported as optimum velocity to abrade the biofilm formation to prevent corrosion. However, in acidic conditions [43], the method to increase the velocity to reduce biofilm formation to prevent corrosion should be reconsidered.

## Conclusion

In this study, the corrosion behaviour of DSS 2205 in SRB environment and control (no bacteria) environment was studied. The presence of SRB in acidic conditions resulted in a lower corrosion rate of material compared to the control environment. It is interesting to note that in such acidic microbial conditions, low velocity could decrease the corrosion process of materials. This is due to the change of environmental pH from acidic to nearly neutral pH caused by SRB metabolic activities. Another factor is the protection of the film including biofilm and corrosion products layer formed materials surface. This could be applied to the transportation industry where metallic pipelines carry fluids such as oil well water where the pH is normally low.

## Supporting information

**S1 Fig. Schematic drawing of experiment.**
(TIF)

**S2 Fig. Electric equivalent circuits for control condition (A) and biotic condition (B).**
(TIF)

## Acknowledgments

The authors acknowledge the support of Charles Darwin University technical staff in preparing experiment.

## Author Contributions

**Conceptualization:** Thi Thuy Tien Tran.

**Data curation:** Thi Thuy Tien Tran.

**Formal analysis:** Thi Thuy Tien Tran.

**Investigation:** Thi Thuy Tien Tran.

**Methodology:** Thi Thuy Tien Tran, Krishnan Kannoorpatti, Anna Padovan, Suresh Thennadil, Khai Nguyen.

**Resources:** Khai Nguyen.

**Supervision:** Krishnan Kannoorpatti, Anna Padovan, Suresh Thennadil.

Writing – **original draft:** Thi Thuy Tien Tran.

Writing – **review & editing:** Thi Thuy Tien Tran, Krishnan Kannoorpatti, Anna Padovan, Suresh Thennadil.

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
