## [Decision Letter · Decision Letter 0]

8 Apr 2021

PONE-D-21-07457

Microbial corrosion of DSS 2205 in an acidic chloride environment under continuous flow

PLOS ONE

Dear Dr. Tran,

Thank you for submitting your manuscript to PLOS ONE. After careful consideration, we feel that it has merit but does not fully meet PLOS ONE’s publication criteria as it currently stands. Therefore, we invite you to submit a revised version of the manuscript that addresses the points raised during the review process.

We look forward to receiving your revised manuscript.

Kind regards,

Mannix Balanay, Ph.D.

Academic Editor

PLOS ONE

Journal Requirements:

Reviewers' comments:

Reviewer's Responses to Questions

**Comments to the Author**

1. Is the manuscript technically sound, and do the data support the conclusions?

Reviewer #1: Yes

Reviewer #2: Yes

2. Has the statistical analysis been performed appropriately and rigorously? 

Reviewer #1: N/A

Reviewer #2: Yes

3. Have the authors made all data underlying the findings in their manuscript fully available?

Reviewer #1: Yes

Reviewer #2: Yes

4. Is the manuscript presented in an intelligible fashion and written in standard English?

Reviewer #1: Yes

Reviewer #2: Yes

5. Review Comments to the Author

Reviewer #1: The manuscript entitled “Microbial corrosion of DSS 2205 in an acidic chloride environment under continuous flow” attempted to assess the influence of bacteria Desulfovibrio vulgaris on the corrosion of stainless steel coupons DSS 2205 at different speeds of seawater flow under microcosm conditions. The manuscript is interesting since the subject has a direct impact on the corrosion of DSS 2205 in a condition that has only been addressed for a few years, despite being a major factor in the formation of corrosive biofilms.

I believe that the manuscript is worthy of being published in the journal PlosOne. I have few considerations:

In the introduction, the citation number 5, although important, is too old, currently, new considerations are addressed in the reviews, so I suggest that they be cited:

Procópio L (2020) The era of 'omics' technologies in the study of microbiologically influenced corrosion. Biotechnol Lett 42 (3): 341-356.

Approaches to the role of corrosion in oil facilities should be cited:

Procópio L (2020) Changes in microbial community in the presence of oil and chemical dispersant and their effects on the corrosion of API 5L steel coupons in a marine-simulated microcosm. Appl Microbiol Biotechnol 104 (14): 6397-6411.

Still in the introduction, in lines 42-44 I suggest citing the publications below. They constitute the few works that have addressed the topic since the manuscript, and should be considered:

Trueba A, Eguía E, Milad MM (2010) Biofouling growth on tubular heat exchangers. Mathematical model and simulation. J Maritime Res 8: 15–34.

Trueba A, García S, Otero FM, Vega LM, Madariaga E (2015) Influence of flow velocity on biofilm growth in a tubular heat exchanger-condenser cooled by seawater. Biofouling 31: 527–534.

Tsai YP (2005) Impact of flow velocity on the dynamic behavior of biofilm bacteria. Biofouling 21: 267-277.

In terms of material and methods, were the analyzes about the coupons used, working as electrode or biofilm analysis, not carried out in triplicate? It is not clear the replication of the experiment as a whole.

I believe that a quantitative estimate of the number of cells present on the coupons would be relevant to validate the differences in biofilms in both conditions. CFU or qPCR techniques can answer this question.

Reviewer #2: The authors investigated the acidic flow for biotic and abiotic corrosion of stainless steel. While the importance of the knowledge of acidic flow is not very clear, the observation shown in the flow system is reasonable, and should be published. I have a few questions regarding the observation.

1. D. vulgaris is anaerobic bacteria and how they could grow under aerobic flow condition?

2. Biofilm formation can be labeled by fluorecent dye, and such experiment should be done.

3. oxygen scavenger conc dependency would be interesting experiments to add.

4. Too many Figure numbers. Some of them can be unified.

6. PLOS authors have the option to publish the peer review history of their article (what does this mean?). If published, this will include your full peer review and any attached files.

Reviewer #1: **Yes: **Luciano Procópio

Reviewer #2: No

---

## [Author Response · Author response to Decision Letter 0]

20 Apr 2021

Response to reviewers’ comments

Dear reviewers,

Thank you for giving us the opportunity to submit a revised draft of our manuscript. We appreciate the time and effort that you have dedicated to providing the valuable feedback on our manuscript. We are grateful to you for your insightful comments on our paper. We have been able to incorporate changes to reflect the suggestions provided by the reviewers. We have highlighted the changes within the manuscript.

Reviewer 1:

The manuscript entitled “Microbial corrosion of DSS 2205 in an acidic chloride environment under continuous flow” attempted to assess the influence of bacteria Desulfovibrio vulgaris on the corrosion of stainless steel coupons DSS 2205 at different speeds of seawater flow under microcosm conditions. The manuscript is interesting since the subject has a direct impact on the corrosion of DSS 2205 in a condition that has only been addressed for a few years, despite being a major factor in the formation of corrosive biofilms.

I believe that the manuscript is worthy of being published in the journal PlosOne. I have few considerations:

In the introduction, the citation number 5, although important, is too old, currently, new considerations are addressed in the reviews, so I suggest that they be cited:

Procópio L (2020) The era of 'omics' technologies in the study of microbiologically influenced corrosion. Biotechnol Lett 42 (3): 341-356.

Response: The recommended reference has been added to the manuscript as reference number 4.

Approaches to the role of corrosion in oil facilities should be cited:

Procópio L (2020) Changes in microbial community in the presence of oil and chemical dispersant and their effects on the corrosion of API 5L steel coupons in a marine-simulated microcosm. Appl Microbiol Biotechnol 104 (14): 6397-6411.

Response: This reference has been added as reference number 5.

Still in the introduction, in lines 42-44 I suggest citing the publications below. They constitute the few works that have addressed the topic since the manuscript, and should be considered:

Trueba A, Eguía E, Milad MM (2010) Biofouling growth on tubular heat exchangers. Mathematical model and simulation. J Maritime Res 8: 15–34.

Trueba A, García S, Otero FM, Vega LM, Madariaga E (2015) Influence of flow velocity on biofilm growth in a tubular heat exchanger-condenser cooled by seawater. Biofouling 31: 527–534.

Tsai YP (2005) Impact of flow velocity on the dynamic behavior of biofilm bacteria. Biofouling 21: 267-277.

Response: These references have been added as references 17-19.

In terms of material and methods, were the analyzes about the coupons used, working as electrode or biofilm analysis, not carried out in triplicate? It is not clear the replication of the experiment as a whole.

Response: The experiments were performed in triplicate. In each experiment, there were two coupons used in the experiment, one coupon was used as working electrode and one coupon was used for biofilm observation. The purpose of using a separate coupon for biofilm observation is to avoid the biofilm damage caused by the applied voltage on the working electrode.

I believe that a quantitative estimate of the number of cells present on the coupons would be relevant to validate the differences in biofilms in both conditions. CFU or qPCR techniques can answer this question.

Response: During the experiments, only the number of bacterial cells in solution were counted using haemocytometer to observe the growth of bacteria and to ensure to have a suitable number of bacteria in the solution (around > 104 cells/mL) (data not shown). The differences in the biofilms were studied qualitatively by observing the coverage of the coupons surfaces by optical microscopy (OM) and scanning electron microscopy (SEM). Unfortunately, we did not perform CFU and qPCR techniques.

 

Reviewer 2: 

The authors investigated the acidic flow for biotic and abiotic corrosion of stainless steel. While the importance of the knowledge of acidic flow is not very clear, the observation shown in the flow system is reasonable, and should be published. I have a few questions regarding the observation.

1. D. vulgaris is anaerobic bacteria and how they could grow under aerobic flow condition?

Response: The experimental work in the manuscript was conducted in a closed-circuit anaerobic condition. The tank, the pipe and the solution were flushed with nitrogen gas before the experiment. The top of the tank was also covered with a layer of nitrogen gas to create an anaerobic condition inside.

3. oxygen scavenger conc dependency would be interesting experiments to add.

 Response: Thank you for your suggestion. We will aim to include such condition in further research to improve the literature.

2. Biofilm formation can be labeled by fluorecent dye, and such experiment should be done.

Response: The experiment was performed in triplicate. In the first experiment, the biofilm formed on the coupon stained with DAPI and observed under fluorescence microscope (photos not shown as low quality) and observed under optical microscope. We found that under OM (optical microscopy), we can see the biofilm formation more clearly and these images were used for the manuscript.

4. Too many Figure numbers. Some of them can be unified.

Response: The number of figures was reduced to 8 figures. Fig. 1 and Fig. 8 were moved to a supplementary file, Fig 2 and 3 were changed to Fig 1(A) and 1(B).

---

## [Decision Letter · Decision Letter 1]

28 Apr 2021

Microbial corrosion of DSS 2205 in an acidic chloride environment under continuous flow

PONE-D-21-07457R1

Dear Dr. Tran,

We’re pleased to inform you that your manuscript has been judged scientifically suitable for publication and but would like to address the authors attention to one of the reviewer's comments to further improve the manuscript.

"The authors should check the concentration of oxygen in the flow system, because oxygen concentration increase in tube and at connections."

Futhermore, it will be formally accepted for publication once it meets all outstanding technical requirements.

Kind regards,

Mannix Balanay, Ph.D.

Academic Editor

PLOS ONE

Additional Editor Comments (optional):

Reviewers' comments:

Reviewer's Responses to Questions

**Comments to the Author**

1. If the authors have adequately addressed your comments raised in a previous round of review and you feel that this manuscript is now acceptable for publication, you may indicate that here to bypass the “Comments to the Author” section, enter your conflict of interest statement in the “Confidential to Editor” section, and submit your "Accept" recommendation.

Reviewer #1: All comments have been addressed

Reviewer #2: All comments have been addressed

2. Is the manuscript technically sound, and do the data support the conclusions?

Reviewer #1: Yes

Reviewer #2: Yes

3. Has the statistical analysis been performed appropriately and rigorously? 

Reviewer #1: Yes

Reviewer #2: Yes

4. Have the authors made all data underlying the findings in their manuscript fully available?

Reviewer #1: Yes

Reviewer #2: Yes

5. Is the manuscript presented in an intelligible fashion and written in standard English?

Reviewer #1: Yes

Reviewer #2: Yes

6. Review Comments to the Author

Reviewer #1: (No Response)

Reviewer #2: Comments were well addressed. However, if possible, authors should check the concentration of oxygen in the flow system, because oxygen concentration increase in tube and at connections.

7. PLOS authors have the option to publish the peer review history of their article (what does this mean?). If published, this will include your full peer review and any attached files.

Reviewer #1: **Yes: **Luciano Procópio

Reviewer #2: No

---

## [Editor Report · Acceptance letter]

30 Apr 2021

PONE-D-21-07457R1 

Microbial corrosion of DSS 2205 in an acidic chloride environment under continuous flow 

Dear Dr. Tran:

I'm pleased to inform you that your manuscript has been deemed suitable for publication in PLOS ONE. Congratulations! Your manuscript is now with our production department. 

Kind regards, 

on behalf of

Dr. Mannix Balanay 

Academic Editor

PLOS ONE